# Evaluation of Candidate Reference Genes for Gene Expression Analysis in Wild *Lamiophlomis rotata*

**DOI:** 10.3390/genes14030573

**Published:** 2023-02-24

**Authors:** Luhao Wang, Feng Qiao, Guigong Geng, Yueheng Lu

**Affiliations:** 1School of Life Sciences, Qinghai Normal University, Xining 810008, China; 2Academy of Plateau Science and Sustainability, Qinghai Normal University, Xining 810008, China; 3Key Laboratory of Tibetan Plateau Medicinal Plant and Animal Resources, Qinghai Normal University, Xining 810008, China; 4Academy of Agricultural and Forestry Sciences, Qinghai University, Xining 810016, China

**Keywords:** *Lamiophlomis rotata*, reference genes, RT-qPCR, geNorm, NormFinder, Bestkeeper, RefFinder

## Abstract

*Lamiophlomis rotata* (Benth.) Kudo is a perennial and unique medicinal plant of the Qinghai–Tibet Plateau. It has the effects of diminishing inflammation, activating blood circulation, removing blood stasis, reducing swelling, and relieving pain. However, thus far, reliable reference gene identifications have not been reported in wild *L. rotata*. In this study, we identified suitable reference genes for the analysis of gene expression related to the medicinal compound synthesis in wild *L. rotata* subjected to five different-altitude habitats. Based on the RNA-Seq data of wild *L. rotata* from five different regions, the stability of 15 candidate internal reference genes was analyzed using geNorm, NormFinder, BestKeeper, and RefFinder. *TFIIS*, *EF-1α*, and *CYP22* were the most suitable internal reference genes in the leaves of *L. rotata* from different regions, while *OBP*, *TFIIS*, and *CYP22* were the optimal reference genes in the roots of *L. rotata*. The reference genes identified here would be very useful for gene expression studies with different tissues in *L. rotata* from different habitats.

## 1. Introduction

Real-time quantitative PCR (RT-qPCR) is an important technology for studying the gene expression pattern in the processes of plant growth and development and secondary metabolite synthesis [1,2,3,4]. It is characterized by high accuracy, strong specificity, and good repeatability [5]. In the process of DNA amplification, RT-qPCR can monitor the change in product content in real time and conduct a quantitative analysis through the change in fluorescence signal intensity, so it is often used for the study of gene expression [6]. However, the accuracy of the results of this technique is affected by various factors, such as the specificity of primers, the integrity of RNA, and the quality of cDNA. Among them, the stability of reference genes has the greatest impact [7]. The selection of appropriate internal reference genes for calibration can reduce the differences between samples and ensure the accuracy of RT-qPCR results [8]. Housekeeping genes have a stable expression and are not easily affected by environment, therefore, they are frequently used as internal reference genes for RT-qPCR analysis. Common housekeeping genes in plants include actin *(ACT)*, 18S ribosomal RNA (*18S rRNA)*, glyceraldehyde-3-phosphate dehydrogenase (*GAPDH*), elongation factor 1α (*EF-1α*), and polyubiquitin (*UBQ*) genes [9]. Given different species, tissues, and experimental conditions, the expression stability of internal reference genes will also be different [10]. For example, the stable expression of *GAPDH* in grape berries is the worst in wheat (*Triticum aestivum*) [11,12]. In the stems of *Psamochloa villosa*, the *ACT* gene is the most suitable internal reference gene for gene expression research under drought conditions, while tubulin alfa chain (*TUA)* is the most stable and the best internal reference gene under cold conditions [13]. Therefore, screening internal reference genes in accordance with different species, tissues, and experimental conditions is necessary to ensure the accuracy of RT-qPCR results [14]. 

*L. rotata* (Benth.) Kudo (*L. rotata*) is a perennial wild herb of the Lamiaceae family, which is widely distributed in the Qinghai, Tibet, Sichuan, and Gansu provinces of China. It grows in meadow, grassland, and sand gravel regions at altitudes between 3100 and 5100 m. It is a commonly used Chinese herbal medicine of Tibetan, Naxi, Mongolian, and other ethnic groups in China [15,16]. The roots and leaves of *L. rotata* contain various active ingredients, such as iridoids, flavonoids, and phenylethanol glycosides [17,18,19]. In clinical practice, this herbal medicine is mainly used to treat traumatic bleeding, bruises, and other diseases [20,21,22]. The iridoid compounds, represented by gardenoside methyl ester and 8-O-acetyl gardenoside methyl ester, which have good hemostatic and procoagulant effects, are the main medicinal components of *L. rotata* [22]. Present research on *L. rotata* mainly focuses on the distribution of germplasm resources [15], the extraction, separation, and identification of chemical components [19], the pharmacological effects [23], and other aspects. However, research on the biosynthetic pathway of iridoid, flavonoid, and phenylethanol glycoside compounds in *L. rotata* has not been reported. It is very important to study the expression pattern of key genes in the biosynthetic pathway of secondary metabolism in wild *L. rotata*, and the research results could provide clues for the molecular mechanism of pharmaceutical ingredient accumulation in *L. rotata*. At present, no research has been conducted on the screening of the internal reference genes of *L. rotata*. Thus, screening the best internal reference gene in the RT-qPCR analysis of *L. rotata* is of great significance for the subsequent study and analysis of the gene expression pattern of *L. rotata*.

With the development of high-throughput sequencing technology, plant RNA-seq datasets can be obtained rapidly and comprehensively [24,25]. This technology is now widely used in transcriptome gene expression analysis [26], secondary metabolism studies of plants [27,28], disease resistance mechanisms [29], biocontrol [30,31], and so on. RNA-seq datasets can also be used to study the molecular function of specific genes and to screen for internal reference genes [32,33]. Therefore, in this study, based on the transcriptome data of leaves and roots in *L. rotata* from five regions (GL, XW, YS, ZD, and CD regions, Figure 1 and Table 1) through the Illumina high-throughput sequencing platform (Biomaker Technologies Co., Ltd., Beijing, China), 15 candidate internal reference genes were finally screened based on the expression of FPKM >10 and difference ploidy < 2 [34], and included *actin 8* (*ACT8*), cyclophilin 22 (*CYP22*), cyclophilin 95 (*CYP95*), TIP 41-like protein (*TIP41*), translation elongation factor (*TFIIS*), *EF-1 α, actin 7* (*ACT7*), protein phosphatase 2A (*PP2A*), oxysterol-binding protein (*OBP*), tubulin β chain (*TUB*), cyclophilin23 (*CYP23*), ribochemical protein large subunit (*RPL*), anthranilate phosphoribosyltransferase (*TrpD*), acetylcholine deacetylase (*AO*), and α colloid NSF attachment protein (*SNAP*). The GeNorm, NormFinder, BestKeeper, and RefFinder algorithms were used to evaluate the expression stability of these candidate genes in the root and leaf tissues of *L. rotata* from different regions. Then, the selected best housekeeping gene could be used to analyze the expression pattern of the genes involved in the biosynthesis pathway of iridoid, flavonoid, and phenylethanol glycoside components of *L. rotata*. The objectives were to further verify the accuracy of the housekeeping gene and lay the foundation for the subsequent research on the functional verification of the genes involved in the biosynthesis pathway of these second metabolism components of *L. rotata*.

## 2. Materials and Methods

### 2.1. Experimental Materials

The wild materials of *L. rotata* used in this research were collected in Guoluo, Xiewu, Yushu, Zaduo, and Chengduo counties in Qinghai Province at different altitudes from 3750 m to 4270 m (Figure 1, Table 1). The root and leaf organs of *L. rotata* from the five areas were analyzed as experimental samples. The samples were placed in liquid nitrogen for rapid freezing and stored in a −80 °C ultra-low-temperature freezer for the following experiments.

### 2.2. Total RNA Extraction and cDNA Synthesis

The total RNA was extracted from the roots and leaves of *L. rotata* from different regions by using a polysaccharide polyphenol plant RNA extraction kit (Nanjing Novozan Biotechnology Co., Ltd., Nanjing, China). The quality of RNA was assessed by 1% agarose gel electrophoresis. RNA was treated with DNase I (New England Biolabs) to remove DNA contamination as per the prescribed protocol. 

The integrity of RNA was determined by observing the brightness of 18S rRNA and 28S rRNA bands through 1.2% agarose gel electrophoresis. The concentration of RNA was detected using a GeneQuant 100 ultraviolet spectrophotometer (Guangzhou Gangran Electromechanical Equipment Co., Ltd., Guangzhou, China), and its purity was determined with OD260/280. Each product used 1000 ng of RNA to synthesize the first strand of cDNA in accordance with the HiScript ^®^ II Q RT SuperMix for qPCR (+gDNA wiper) kit (Nanjing Novozan Biotechnology Co., Ltd., Nanjing, China), and the cDNA was stored in a −20 °C low-temperature freezer for standby.

### 2.3. Screening of Candidate Internal Reference Genes and Designing of Primers

Based on the transcriptome data of the roots and leaves of *L. rotata* from different regions, 15 candidate internal reference genes, including *ACT8*, *CYP22*, *CYP95*, *TIP41*, *TFIIS*, *EF-1α*, *ACT7*, *PP2A*, *OBP*, *TUB*, *CYP23*, *RPL*, *TrpD*, *AO*, and *SNAP*, were finally screened. In accordance with the nucleotide sequence of candidate internal reference genes, primers were designed using Primer Premier 5.0 software (Table 2). The parameters were set as follows: GC content of 45–65%, primer length of 21–27 bp, primer annealing temperature of 60–67 °C, and amplification length of 80–200 bp. The primer was synthesized by Beijing Aoke Dingsheng Biotechnology Co., Ltd. (Beijing, China).

### 2.4. RT-qPCR Analysis of the Candidate Internal Reference Genes

The original solution of cDNA extracted from the root and leaf organs of *L. rotata* from different regions and diluted 5 times was taken as the template. The QuantStudio™ 6 Flex System (Thermo Fisher Scientific, Waltham, MA, USA) was used to conduct RT-qPCR detection. The reaction system had the following components: 10 μL of 2 × ChamQ Universal SYBR qPCR Master Mix, 7.2 μL of ddH_2_O, 2 μL of cDNA, and 0.4 μL of upstream and downstream primers (10 μM) each. The reaction conditions were as follows: predenaturation at 95 °C for 30 s, 95 °C for 10 s, and 60 °C for 30 s; 40 cycles. After amplification, the 60–95 °C dissolution curve was analyzed to detect the specificity of the primers. 

### 2.5. Determination of the Correlation Coefficient of Primer Pairs 

The same amount of cDNA template stock solution of samples from different parts was mixed and diluted 5 times in turn. Five concentration gradients of the template stock solution were set, which were 5, 5^−1^, 5^−2^, 5^−3^, and 5^−4^. Each serial dilution was used separately in RT-qPCR to determine the threshold cycle (Ct) values of the candidate reference genes. 

The primer specificity was verified by the presence of a single peak in the melt curve analysis during the RT-qPCR process. Three independent biological replicates and three technical repetitions were performed for each of the quantitative PCR experiments. The threshold cycle (Ct) was measured automatically, and correlation coefficients (R^2^), together with slope, were calculated from the standard curve based on a five-fold series dilution of the cDNA templates. The corresponding RT-qPCR efficiencies (E) for each gene were determined from the given slope. 

### 2.6. Stability Analysis of the Candidate Internal Reference Genes

GeNorm [35], NormFinder [36], BestKeeper [37], and RefFinder [38], four conventional software tools, were used for evaluating the stability of the candidate reference genes under various altitude habitats. The Ct values were changed to relative quantities (Q) with the formula Q = 2^−△Ct^, where △Ct = each corresponding Ct − minimum Ct. Then, the Q-values were input into GeNorm and NormFinder to calculate stability values (SVs). GeNorm uses the average stability value (M value) to evaluate the expression stability, which involves the average pairwise variation between each reference gene and others. Genes with *M* < 1.5 are generally regarded as stable. Moreover, the GeNorm pairwise variation (V) values (Vn/Vn + 1) were applied to define the most appropriate reference gene number. The recommended value is ≤0.15 when opting for an appropriate number of genes. The NormFinder software evaluates expression stability by calculating the SV (lower SV indicates higher stability) when applying reference genes for standardization. In BestKeeper, the most stable reference gene was evaluated by the coefficient of variance (CV) and standard deviation (SD), with lower CV and SD values meaning higher stability. The RefFinder tool (http://blooge.cn/RefFinder/ accessed on 1–2 November 2022), integrating the outcomes of BestKeeper, Delta Ct, GeNorm, and NormFinder analysis, was applied to perform the most stable reference genes.

### 2.7. Validation of the Validity of the Candidate Reference Genes 

When the selected reference gene was used as internal reference, the CT values of three genes encoding key enzymes (deoxyxylulose 5-phosphate synthase (*DXS*, EC 2.2.1.7), deoxyxylulose 5-phosphate reductoisomerase (*DXR*, EC1.1.1.267), and 4-Hydroxy-3-methylbut-2-en-1-yl diphosphate reductase (*HDR*, EC 1.17.1.2)) related to the biosynthesis of terpenoids were determined by RT-qPCR. The obtained Ct values were sorted out and analyzed quantitatively with 2^−△△Ct^ to verify the reliability of the candidate reference gene [39]. The primers of each target gene were designed by Primer Premier 5.0 software and shown in Table 3.

## 3. Results

### 3.1. Primer Specificity and Correlation Coefficient of Primer Sets 

Fifteen candidate reference genes were selected from the transcriptome of *L. rotata*. The specificity of each primer pair was assessed by agarose gel electrophoresis and melting curve analysis (Appendix A). For each candidate reference gene, the length of a single amplicon ranged from 86 bp to 197 bp (Table 2). 

To check the efficiency of PCR amplification using these primers, the correlation coefficient (R^2^) of each primer set was determined using the slopes of the curves obtained by serial dilutions. For this purpose, a dilution series of template samples containing cDNA and water in the ratio of 1:1, 1:5, 1:25, 1:125, and 1:625 was prepared, and RT-qPCR was performed (Figure 2). The plot of the Ct values versus the natural logarithm (ln) of cDNA dilutions demonstrated a linear distribution for all primer sets (Figure 2). The calculation of R^2^ indicated that all primer sets had R^2^ values more than 0.95 (Figure 2). R^2^ values between 0.80 and 1.00 are considered very good for any primer set. The amplification efficiency (E) of the primers was between 90% and 114% (Table 2). Based on the values of E and R^2^, we concluded that the primer specificity of each candidate reference gene was strong and thus can be used for screening candidate reference genes.

### 3.2. Ct Values of the Candidate Reference Genes 

Ct value can reflect the expression abundance of candidate internal reference genes in different samples. The Ct value of the 15 candidate internal reference genes in all samples was between 21 and 34. The Ct value across all the samples was shown in the form of a box and whisker plot (Figure 3). The median of the Ct values of the candidate reference genes ranged from 24.57 (for *EF*-*1α*) to 32.33 (for *TrpD*). Lower Ct values implied higher expression levels and vice versa. The box and whisker plot indicated that *EF*-*1α* and *ACT7* had the highest levels among all the candidate reference genes examined. With the median of the Ct value as the evaluation standard, *EF-1α* had the highest expression level with a median of 24.57. The expression levels of *ACT7* and *OBP* were also relatively high, with median values of 25.69 and 26.75, respectively. Interestingly, genes with the lowest expression levels (*TrpD* and *SNAP*) showed smaller variability in their Ct values across different regions with median values of 32.23 and 32.11. 

Similarly, Figure 3 shows that the upper and lower box lines and the maximum and minimum values of *CYP95* and *TFIIS* varied minimally in all samples. The expression of these two genes was preliminarily believed to be relatively stable in all samples. On the contrary, the difference between the upper and lower quartiles and the maximum and minimum values of *TUB* was relatively large. The expression stability of this gene was speculated to be poor in all samples. Meanwhile, the difference of *RPL* and *ACT8* was small in leaves from different regions, which suggested that these two genes were relatively stable. In the root of *L. rotata* from different regions, the difference of *OBP* and *PP2A* was small, and these two genes were considered to be relatively stable.

### 3.3. Stability Analysis of the Candidate Reference Genes

To systematically evaluate the stability of the 15 candidate reference genes, this study used GeNorm, NormFinder, BestKeeper software, and the RefFinder online analysis tool to analyze the results of RT-qPCR.

#### 3.3.1. GeNorm Analysis of the Stability of the Candidate Reference Genes

The stability of the 15 candidate reference genes was analyzed using GeNorm software. The stability was expressed by M value: the smaller the M value, the stronger the stability. The M value of the 15 candidate reference genes in each group was less than 1.5 (Figure 4), which indicated that the candidate reference genes had good stability. In all samples, the M value of *CYP22/TFIIS* was the smallest (0.54), whereas the M value of *TUB* was the largest (1.18). Accordingly, the expression stability of *CYP22/TFIIS* was the best in all samples, and it was the most suitable to be used as an internal reference gene. For the leaves with unique characteristics from different regions, the M values of the 15 candidate reference genes were ranked from largest to smallest as *TUB* > *AO* > *TrpD* > *OBP* > *ACT7* > *CYP23* > *SNAP* > *TIP41* > *CYP95* > *ACT8* > *RPL* > *PP2A* > *CYP22* > *TFIIS/EF-1α*. However, in the roots of *L. rotata* from different regions, the M value of *TFIIS/OBP* was the smallest (0.25), which was greatly different from the stability evaluation results of all samples. The M value of *OBP* in all samples was speculated to be higher because of the poor expression stability in the leaves of *L. rotata* from different regions.

Through the GeNorm software, the 15 candidate reference genes were analyzed for the difference in candidate reference gene standardization factor pairs. The results (Figure 5) showed that V2/V3 was greater than 0.15 and V3/V4 was less than 0.15 in all samples and leaves from different regions. Therefore, three reference genes should be introduced in the RT-qPCR analysis of these two groups to make the results accurate and reliable. However, in the roots of *L. rotata* from different regions, V2/V3 (0.113) < 0.15, which indicated that the two internal reference genes can make the analysis accurate and reliable when RT-qPCR analysis was carried out in this group.

#### 3.3.2. NormFinder Analysis of the Stability of the Candidate Reference Genes

The stability of the 15 candidate reference genes was analyzed using NormFinder software. Similar to GeNorm, the lower the value is, the stronger the stability of genes is. In Figure 6, the stability sequencing results of different candidate reference genes in three different tissues were listed from left to right. In leaves from different regions, *TFIIS* had the best stability (0.22), whereas *TUB* had the worst stability (1.43). In the roots of *L. rotata* from different regions, the stability of *OBP* was the best (0.13), whereas the stability of *TrpD* was the worst (0.67). *CYP22* and *TFIIS* were stable in the top four in all experimental conditions and could be considered the best candidate reference genes. By contrast, *TUB* and *TrpD* had poor stability under all experimental conditions.

#### 3.3.3. BestKeeper Analysis of the Stability of the Candidate Reference Genes

BestKeeper is an Excel-based algorithm, which directly uses the Ct value obtained from RT-qPCR analysis for further analysis. BestKeeper evaluates the stability of the candidate reference gene by calculating its CV and SD. Smaller CV and SD values represent better stability. From Table 4, in the leaves the stability of *OBP* (6.57 ± 1.86) was the worst, whereas that of *TrpD* (2.84 ± 0.92) and *CYP95* (3.51 ± 1.07) was the best In the roots from different regions, the stability of *ACT7* (4.39 ± 1.08) was the worst, whereas that of *TrpD* (1.58 ± 0.51) and *AO* (1.65 ± 0.50) was the best. Under all experimental conditions, the stability of *TrpD* ranked first, and it was considered the most suitable reference gene.

#### 3.3.4. RefFinder Analysis of the Stability of the Candidate Reference Genes

The RefFinder online analysis tool can take appropriate weight to comprehensively evaluate the analysis results of GeNorm, NormFinder, and BestKeeper, avoid the one-sidedness of single program analysis, and better analyze candidate internal reference genes [23]. In this study, 15 candidate internal reference genes were analyzed by RefFinder (Figure 7), and the results showed that the most suitable internal reference genes were *TFIIS*, *EF-1α*, and *CYP22* in the leaves of *L. rotata* from different regions. The most suitable internal reference group in the roots of *L. rotata* from different regions included *OBP*, *TFIIS*, and *CYP22*. In all samples, the most suitable internal reference group was *TFIIS*, *CYP95*, and *CYP22* based on the analysis of RefFinder software.

### 3.4. Verification of the Stability of Internal Reference Genes

Three relatively stable internal reference genes (*TFIIS*, *EF-1α*, and *CYP22*), and a poorly stable internal reference gene (*TIP41*) from the leaf, were used as the internal reference to study the expression of three genes (*DXR*, *HDR*, and *DXS*) related to the synthesis of iridoid compounds in the leaves of *L. rotata* from different regions. As shown in Figure 8, the three genes *DXR*, *HDR*, and *DXS* all showed the same expression change trend in the leaves of *L. rotata*.

Three stable internal reference genes (*TFIIS*, *OBP*, and *CYP22*) were also used to study the expression of the three genes *DXR*, *HDR*, and *DXS* in roots (Figure 8). Based on the two internal reference genes of *TFIIS* and *OBP*, the specific genes *DXR*, *HDR*, and *DXS* all showed the same expression change trend in the roots of *L. rotata* from five habitats.

Meanwhile, in the different tissues of *L. rotata* from different regions, different reference genes were screened as the most optimal internal reference genes. The three genes *TFIIS*, *EF-1α*, and *CYP22* were found to be the three most stable reference genes in leaves (Table 4 and Figure 6, Figure 7 and Figure 8). In roots, the three genes *TFIIS*, *OBP*, and *CYP22* were the most stable reference genes (Table 4 and Figure 6, Figure 7 and Figure 8). 

## 4. Discussion

RT-qPCR is a common technical means for studying gene expression patterns and analyzing gene functions and molecular mechanisms in molecular biology research. Appropriate reference genes are the prerequisite to ensure the accuracy of RT-qPCR data. In recent years, many scholars have analyzed the reference genes of different kinds of plants based on the RT-qPCR technology and found that the internal reference genes are not universal under the conditions of different plants, the same plant from different sources, and different parts of the same plant, and that the most suitable reference genes are different [40]. For example, in different tissues of *Isatis indigotica*, *CYP* was the most stable reference gene [41], but in all samples of *Lycoris aurea*, the stability of the *CYP* gene was poor [42]. The most stably expressed reference genes in the roots, stems, and leaves under high-light stress were *PSKS1*, *UBC2*, and *PSKS1*, respectively [43]. The *EF-1α* gene was identified as an unstable reference gene in response to herbicide stress in wild oat [44]. In all samples or different tissues of jute, the stability of *PP2A* gene expression was the best. On the contrary, in jute under drought stress, the stability of *PP2A* was poor, and the stability of the *ACT7* gene was the strongest [45]. The *ACT*, *RIB*, and *TUA* genes were the most stable reference genes expressed in different tissues of *Schima superba* [46]. Therefore, studying candidate reference genes from different sources and different parts is of great significance for improving the reliability of RT-qPCR results. 

The identification of the optimal reference genes for specific conditions in a given species is imperative. Furthermore, our study revealed that the stability ranking of reference genes varied in certain circumstances due to the different algorithms used in the three analytical tools. In this study, the stability of candidate reference genes was evaluated for the first time for different regions and different parts of wild *L. rotata*. A total of 15 candidate reference genes were selected, and the expression stability of these 15 candidate reference genes in the roots and leaves of *L. rotata* from different regions was analyzed through GeNorm, NormFinder, and BestKeeper software. The results of the different software were generally consistent, but there were still differences. For example, based on the analysis of GeNorm, *TFIIS* and *EF-1α* were the best (Figure 4). The NormFinder analysis showed that the expression level of *TFIIS* was the most stable (Figure 6). Under the analysis of BestKeeper, *CYP95* and *TrpD* had the lowest degree of variation and the best stability. The differences were due to the fact that the algorithms of the three software are different; similar findings have been obtained in previous studies [46,47]. 

In order to screen the most suitable reference genes, the results obtained from the above three software programs were comprehensively analyzed using the online analysis tool RefFinder. The results showed that *TFIIS* and *EF-1α* were the most stable genes in the leaves of *L. rotata* in different regions based on the RefFinder method. The *TFIIS* gene is an important multifunctional protein, which can participate in gene translation regulation, cell signal pathway, and skeleton composition in organisms, and plays an important role in cell life activities. It is a common reference gene in plants at present [48]. *EF-1α* is a ubiquitous and conserved cytosolic protein among eukaryotic organisms and is responsible for catalyzing the binding of aminoacyl-transfer RNAs to ribosomes [49,50]. The expression of *OBP* and *TFIIS* was most stable in the roots of *L. rotata* in different regions based on the RefFinder method. The oxysterol-binding protein (*OBP*) is a storied protein in organelle biology and in sterol signaling and/or sterol transport functions [51]. Its major roles include acting as a membrane contact site tether, as well as a lipid antiporter [52]. Under Fe-deficient conditions, *OsOBP* and *OsLUG* were found to be the two most stable reference genes in any type of tissue taken in the study [53]. Among all the samples from different regions of *L. rotata*, *TFIIS* and *CYP95* were the most stable reference genes based on the RefFinder method. Plant cyclophilins (*CYPs*) are widely involved in a range of biological processes, including stress response, metabolic regulation, and growth and development [54,55]. *CYP* belongs to the immunophilin family and has peptidyl-prolyl *cis*-*trans* isomerase (PPIase) activity, which catalyzes the *cis*-*trans* isomerization process of proline residues. *CypA* demonstrated the most consistent expression irrespective of disease severity and emerged as the most suitable reference gene in COVID−19 and CAM [56]. 

In order to further verify the selected internal reference genes from the screen, we selected *DXR*, *DXS*, and *HDR* as the target genes. *DXR*, *DXS*, and *HDR* are key functional genes in the biosynthetic pathway of iridoid compounds and play roles in regulating the accumulation of downstream products and influencing the biosynthesis of iridoid compounds [57,58,59]. For example, the overexpression of the *DXR* gene induced by MeJ in *Tripterygium wilfordii* can increase the accumulation of triptophenolide [60]. Overexpression of the *DXS2* gene in the hairy root of *Salvia miltiorrhiza* will increase the content of tanshinone, while down-regulation of the *DXS2* gene will significantly reduce the content of tanshinone [61]. Overexpression of the *HDR* gene in *Nicotiana tabacum* will cause the expression of downstream genes related to terpene biosynthesis to be up-regulated, thus causing a significant increase in the accumulation of terpene compounds [62]. In this study, the expression levels of three target genes mentioned above were analyzed using the three most stable internal reference genes and one less stable internal reference gene. The results showed that the expression pattern of a same target gene (*DXR*, *DXS*, or *HDR)* was very similar although using the three most stable reference genes (*TFIIS, EF-1α*, or *CYP22* in leaves) (Figure 8), while the expression pattern of a same target gene (*DXR*, *DXS*, or *HDR)* changed more or less using a less internal reference gene (*TIP41*) (Figure 8). The validation results further illustrated the reliability of the ReFinder analysis results, and our study results also showed the importance of selecting the appropriate internal reference genes to ensure the reliability of RT-qPCR results.

## 5. Conclusions and Future Perspectives

The best genes for normalization of RT-qPCR in different tissues of *L. rotata* from different habitats are different. The three genes *TFIIS*, *EF-1α*, and *CYP22* are stable, viable alternatives for data normalization in the leaves of *L. rotata*. Meanwhile, the genes *TFII*, *OBP*, and *CYP22* are the most stable reference genes in the roots of *L. rotata*. Our results reinforce the importance of the determination of the transcription stability of different candidate genes considering each experimental condition of interest. Our findings establish a solid foundation for further study of the gene regulatory network of *L. rotata* in response to different-altitude habitats. What is more, we can use the best reference genes screened from the leaves and roots of the wild *L. rotata* herb to analyze the expression changes of the genes related to the accumulation of the main medicinal compounds (iridoids, flavonoids, and phenylethanosides) of *L. rotata* and further explore the molecular regulation mechanism of the accumulation of the medicinal components of *L. rotata* at different altitudes. 

## Figures and Tables

**Figure 1 genes-14-00573-f001:**
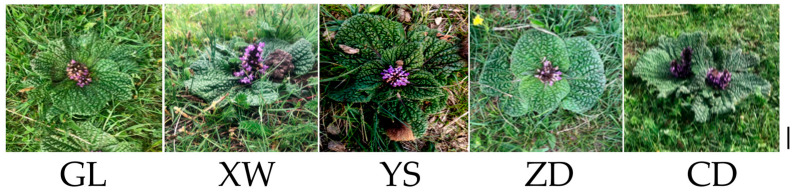
*L. rotata* in five different habitats. GL: Guoluo county in Qinghai province; XW: Xiewu county in Qinghai province; YS: Yushu county in Qinghai province; ZD: Zaduo county in Qinghai province; CD: Chengduo county in Qinghai province. Bar = 2 cm.

**Figure 2 genes-14-00573-f002:**
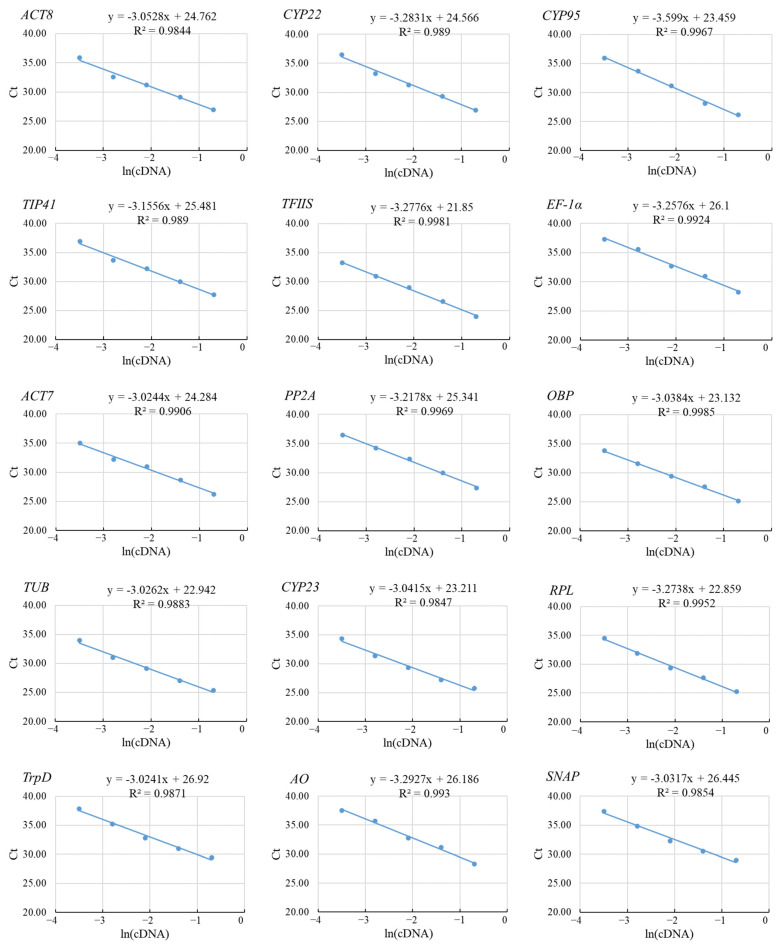
The correlation coefficient of each primer pair used for RT-qPCR. A graph of Ct value vs. the serial dilution factor of cDNA was drawn to determine the correlation coefficient (R^2^). Its value for each reference gene has been mentioned within the respective graph.

**Figure 3 genes-14-00573-f003:**
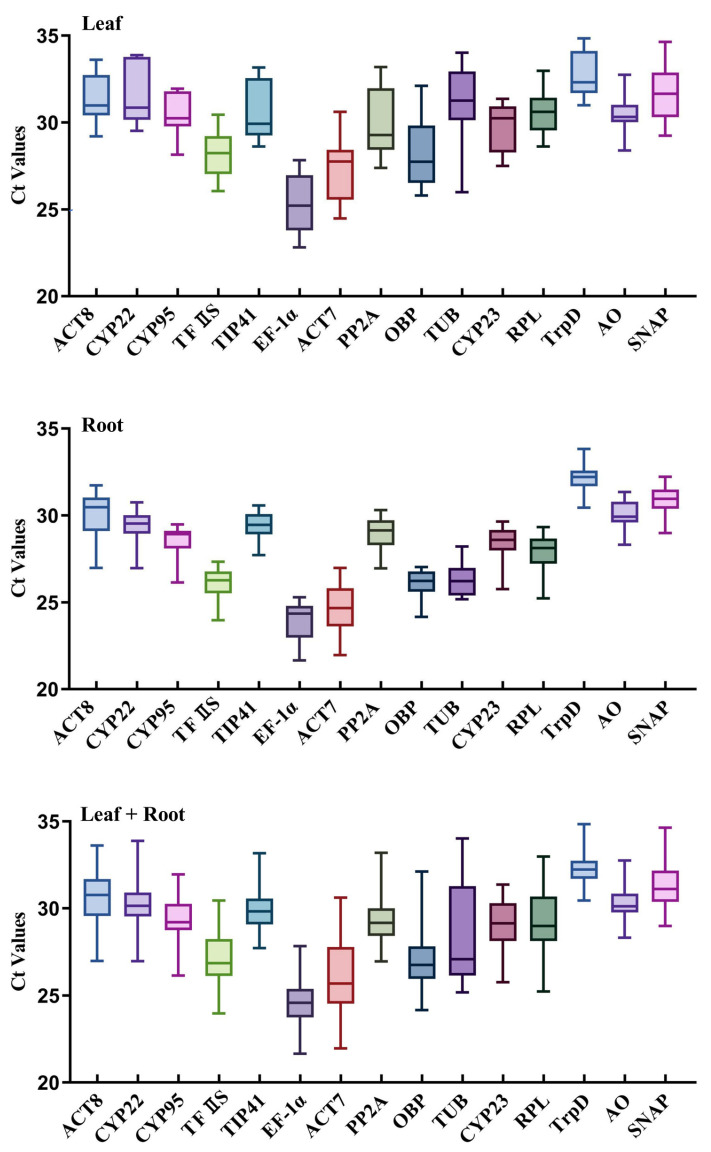
Gene expression levels of the candidate reference genes displayed as Ct values across leaf, root, and leaf + root samples of *L. rotata* from five habitats. The graph also shows the variation in Ct values for each reference gene. Whiskers across the box depict the highest and lowest values.

**Figure 4 genes-14-00573-f004:**
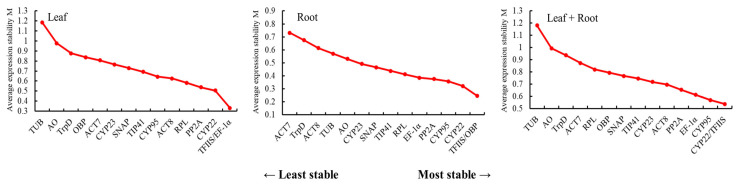
Average expression stability (M value) of fifteen candidate reference genes based on geNorm analyses. The most stably expressed reference genes have the lowest M values. M values were calculated for leaf, root, and leaf + root tissues of *L. rotata* from five habitats. Genes with M values lower than 1.5 may be used as candidate reference genes.

**Figure 5 genes-14-00573-f005:**
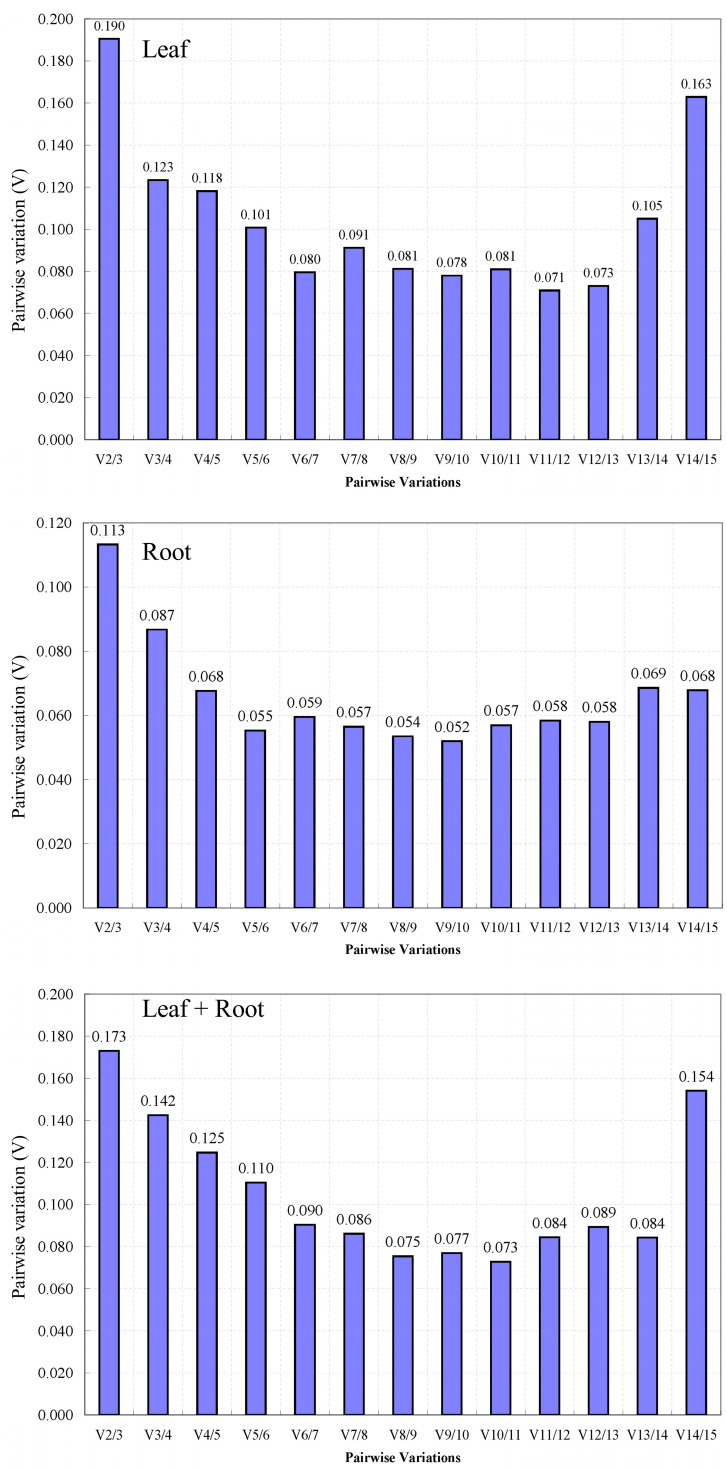
Determination of the optimal number of reference genes for each type of different habitat through geNorm analysis. Pairwise variation (V) was calculated by geNorm to determine the required number of reference genes for accurate normalization under different habitats. geNorm calculates pairwise variation (Vn/Vn + 1) for the normalization factors NFn and NFn + 1 to determine (V < 0.15) the optimal number of reference genes.

**Figure 6 genes-14-00573-f006:**
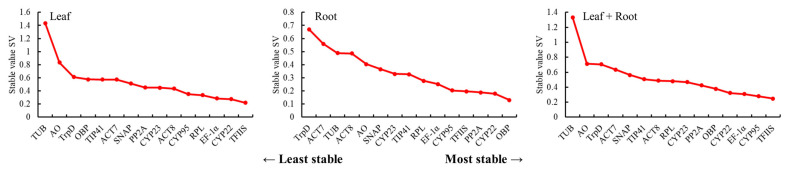
Expression stability values for 15 candidate reference genes calculated by NormFinder.

**Figure 7 genes-14-00573-f007:**
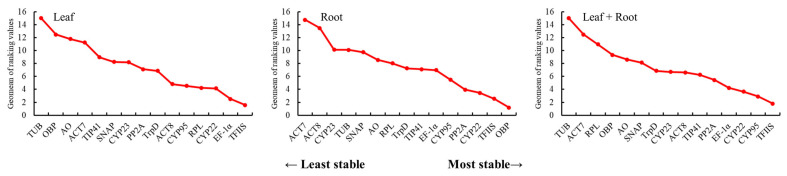
Ranking of candidate reference gene stability by RefFinder analysis.

**Figure 8 genes-14-00573-f008:**
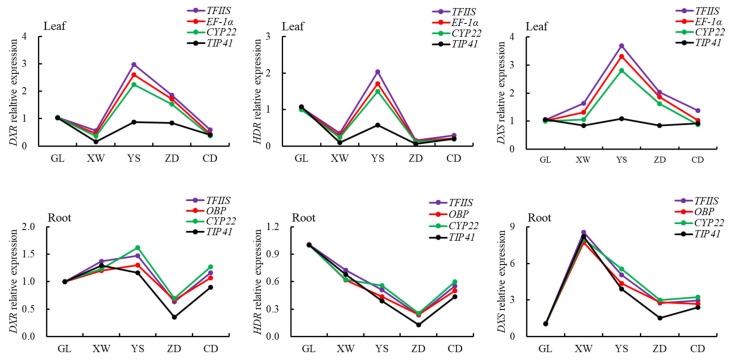
The relative expression of *DXR*, *HDR*, and *DXS* of *L. rotata* roots and leaves in different regions. *TFIIS*, *EF-1α*, *CYP22*, *OBP*, and *TIP41* all were reference genes. *DXR, HDR*, and *DXR* all were target genes related to the biosynthetic pathway of iridoid compounds in *L. rotata*. The purple curve was based on the *TFIIS* reference gene, the red curve was based on the *EF-1α* reference gene in leaves *or OBP* reference gene in roots, the green curve was based on the *CYP22* reference gene, and the black curve was based on the *TIP41* reference gene.

**Table 1 genes-14-00573-t001:** The habitat and site of *L. rotata* from five regions.

Sample Name	Sampling Location	Altitude	East Longitude	North Latitude
GL	Guoluo county, Qinghai, China	3750 m	100°14′38″	34°29′10″
XW	Xiewu county, Qinghai, China	3860 m	97°21′3″	33°7′46″
YS	Yushu county, Qinghai, China	3880 m	97°1′23″	32°51′4″
ZD	Zaduo county, Qinghai, China	4208 m	95°10′48″	32°52′12″
CD	Chengduo county, Qinghai, China	4270 m	97°27′16″	33°18′2″

**Table 2 genes-14-00573-t002:** Primer sequence information for reference genes in RT-qPCR analyses.

Gene Name	Gene Transcript Genbank No.	Primer Sequence	Product Length (bp)	Primer Efficiency
*ACT8*	OQ471970	F: ACTCACTTGCGGTCCAGTTATCC R: ATAACAGCTCCAGGGACTTCCAC	107	1.13
*CYP22*	OQ471972	F: CTTCGACATCACCATCGGAAAC R: TTCCTGTATTCGCCTGTGCAGT	117	1.02
*CYP95*	OQ471971	F: GCAACGGTCTCTCCTCCAAGA R: CTCACAGGGCTTCGACTTGGT	86	0.90
*TIP41*	OQ471973	F: GGGAAGACTGCCAGGATCAAAT R: AGCCAGAAGCGCAAGAGAAGAT	175	1.07
*TFIIS*	OQ471974	F: GAGTTTGAGCCACGCTCGATT R: TCTTGCACCTCCCACAAGTGA	163	1.02
*EF-1α*	OQ471975	F: ACTGGGACTTCTCAGGCTGATTG R: CTGGCCTTGGAGTACTTTGGTGT	188	1.03
*ACT7*	OQ471976	F: CACCACCCGAAAGAAAGTACAGTG R: AGGACCCGATTCATCATACTCTCC	110	1.14
*PP2A*	OQ471977	F: GCTCATGTGTTGCTCCCTCCT R: GCTGCCAGCCTCTTCACAAGT	155	1.05
*OBP*	OQ471978	F: GGAACCTCTTCCTGGCACAGA R: AACCAGACTTGGCGAGGTCAC	197	1.13
*TUB*	OQ471979	F: CAAACTCGCCGTGAACCTCAT R: CGTCCCACATTTGCTGGGTAA	134	1.14
*CYP23*	OQ471980	F: TGCGCCCTGTGCAATTCTATC R: AACTGTCTTTGGCGCGACACT	132	1.13
*RPL*	OQ471981	F: GAAACCCGCTGTCGTTAAACC R: CTATCATCGCGCTTTCCTTCC	151	1.02
*TrpD*	OQ471982	F: CTGAGGCTGAGGCTTCTCTTGA R: CACCACCAGTCCCAACAATGTC	192	1.14
*AO*	OQ471983	F: GTCGCCTACAAGCCAAATAGGG R: GACGACATCCATGTGCATACCA	99	1.01
*SNAP*	OQ471984	F: GTATGAAGACGCTGCCGATTTG R: GCATTAGCTGCTTCATGCTTGC	141	1.14

**Table 3 genes-14-00573-t003:** Primer sequence information for target genes in *L. rotata*.

Gene Name	Gene Transcript Genbank No.	Primer Sequence (5′–3′) (Forward/Reverse)	Amplicon Length (bp)
*DXS*	OQ471985	F: GAAGGGGAGAGGGTGGCTCTAT R: GAGCACCATCCAACGGCTTAC	136
*HDR*	OQ471986	F: CTTGCCGGAGACCAGAATATC R: GCCTTGGCGTTAAACTCAGAC	111
*DXR*	OQ471987	F: CGAGCAGAACTTGTCACATCG R: CTGCAAACTAGCCGCGTAATC	84

**Table 4 genes-14-00573-t004:** Ranking of candidate reference gene stability by BestKeeper analysis.

Reference Gene	Leaf			Root			Leaf + Root		
	CV Value	SD Value	Rank	CV Value	SD Value	Rank	CV Value	SD Value	Rank
*TrpD*	2.84	0.92	1	1.58	0.51	1	2.28	0.74	1
*AO*	4.09	1.27	6	1.65	0.50	2	2.79	0.85	2
*SNAP*	4.16	1.32	7	2.19	0.68	8	3.35	1.05	3
*CYP95*	3.51	1.07	2	2.39	0.68	10	3.39	1.00	4
*ACT8*	3.52	1.10	3	3.87	1.16	14	3.54	1.09	5
*CYP23*	3.86	1.15	5	2.90	0.82	12	3.58	1.04	6
*TIP41*	4.81	1.48	9	2.06	0.61	6	3.59	1.08	7
*PP2A*	5.40	1.62	11	1.88	0.55	3	3.73	1.10	8
*CYP22*	4.88	1.55	10	2.12	0.62	7	4.35	1.33	9
*EF-1α*	5.78	1.47	13	3.05	0.74	13	4.45	1.10	10
*TFIIS*	4.44	1.25	8	2.24	0.58	9	4.48	1.22	11
*RPL*	3.58	1.10	4	2.78	0.77	11	5.15	1.51	12
*OBP*	6.57	1.86	15	2.05	0.53	5	5.24	1.43	13
*ACT7*	5.51	1.51	12	4.39	1.08	15	6.46	1.68	14
*TUB*	6.02	1.86	14	2.01	0.53	4	9.35	2.68	15

## Data Availability

The data are available at https://www.ncbi.nlm.nih.gov/Genbank/update.html, GenBank accession numbers OQ471970; OQ471971; OQ471972; OQ471973; OQ471974; OQ471975; OQ471976; OQ471977; OQ471978; OQ471979; OQ471980; OQ471981; OQ471982; Q471983; OQ471984; OQ471985; OQ471986; OQ471987.

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
