# Peer review of "Evaluation of Candidate Reference Genes for Gene Expression Analysis in Wild Lamiophlomis rotata"

_genes, 2023, doi:10.3390/genes14030573_

Round 1
Reviewer 1 Report
I appreciate the opportunity to help revise this manuscript (Evaluation of Candidate Reference Genes for Gene Expression Analysis in Wild Lamiophlomis rotata as a Chinese and Tibetan Medicine Plant from Different-Altitude Habitats).
Some remarks are yellow in the text.

Author Response
Reply to the reviewer 1 Point 1: Title in my opinion could be more succinct. Response 1: Title was changed and named “Evaluation of Candidate Reference Genes for Gene Expression Analysis in Wild Lamiophlomis rotata” Point 2: Do not use the same words that are mentioned in the title. Response 2: Key words were modified. Keywords: Lamiophlomis rotata; reference genes; RT-qPCR, geNorm; NormFinder; bestkeeper; RefFinder Point 3: Intended readers, and some readers with a certain degree of myopia, please adjust the 4 colors of the figures. Response 3: The figures all were enlarged. Point 4: I think it is feasible to cite manuscripts that describe the selection of reference genes in medicinal plants. Response 4: Add 17 references, including some articles related to medicinal plants. Point 5: Enter date information, ICG (http://icg.big.ac.cn) Response 5: We enter the ICG (http://icg.big.ac.cn) and read some information. Point 6: refrigerator? Response 7: Change refrigerator into ultra low temperature freezer. Point 7: Self-complementary and hair-pin structures were avoided?how did you avoid this problem? Response 7: At least two-pair primers of one reference gene were designed and used for RT-qPCR. The specificity of each primer pair was assessed by agarose gel electrophoresis and melting curve analysis. Finally, one-pair proper primers was screened with one band by agarose gel electrophoresis and one peak by melting curve analysis, which avoid the self-complementary and hair-pin structures. Point 8: Better describe the data analysis to evaluate the expression stability of the reference genes. Response 8: In lines 157-173, all sentences were organized and described again. Point 9: If this is your first time, mention in full: Response 9: Add the full name in lines 176-178. Point 10: Standardize the figure according to the journal's guidelines. Response 10: Add the bar and standardize Figure 1. Point 11: Improve figure layout and centering. Response 11: Figures were improved and enlarged. Point 12: This table 1 could be add in Supplemental Information. Response 12: Add “Qinghai, China” information in Table 1. Point 13: Standardize the table according to the journal's guidelines. Response 13: Standardize the table according to the journal's guidelines. Point 14: This figure 8 could be add in Supplemental Information. Add in bold according to norms. Response 14: Add supplemental information in Figure 8. Point 15: Mention the different regions in the figure8. Rewrite legend. Response 15: Add different color curve information in lines 378-382 of Figure 8. Point 16: Place the manuscript according to the journal's guidelines. Response 16: Correct and modify the paper structure and writing. Point 17: I suggest authors to make a better discussion based on their data. Response 17: Modify the discussion in lines 426-433 and add the last paragraph in lines 440-458. Point 18: Authors could place: 6. Conclusions and Future Perspectives and additinally, place a setence about perspectives Response 18: Modify the conclusions and future perspectives in line 460 and add a sentence about perspectives in lines 469-473. Point 19: Detail the legend better so that it is didactic for the reader.(figures1) Response 19: Supplement the information in lines 479-481.Reviewer 2 Report
In this study, authors determined the reference genes for root and leaf tissues of Lamiophlomis rotata plants growing in various habitats. The study includes RT-PCR experiments with diluted cDNA templates and results analyzed by statistical methods. However, MS requires some explanations indicated below:
In the abstract “Based on the RNA-Seq data of wild L. 20 rotata from five different regions, the stability of 15 candidate internal reference genes were analyzed 21 using geNorm, NormFinder, and BestKeeper.”
However, the transcriptome data was not given. The role of RNA-Seq should be clarified throughout the manuscript. It is not clear which transcriptome data they are referring to. If it is already published data it needs to be cited. If it is unpublished data they should give more information about it. Also, they should give more information about how they used the transcriptome data to select the genes.
The authors used geNorm, NormFinder, BestKeeper, and RefFinder, however, they do not mention about RefFinder in Lines 22, 145, and 373, and they talk about 3 analytical tools in Lines 369-379. Please provide results for all the tools used in the study.
In Line 148, does “minimum Ct value” mean “median Ct value”?
Under title 4.1, various literature were given however, they are not related to the existing study. This title can be removed completely. The Discussion section should be completely rewritten.
Author Response
Author's Reply to the Review Report (Reviewer 2) In this study, authors determined the reference genes for root and leaf tissues of Lamiophlomis rotata plants growing in various habitats. The study includes RT-PCR experiments with diluted cDNA templates and results analyzed by statistical methods. However, MS requires some explanations indicated below: Point 1: In the abstract “Based on the RNA-Seq data of wild L. rotata from five different regions, the stability of 15 candidate internal reference genes were analyzed using geNorm, NormFinder, and BestKeeper.” Reponse 1: Based on the RNA-Seq data of wild L. rotata from five different regions, the stability of 15 candidate internal reference genes was analyzed using geNorm, NormFinder, BestKeeper and RefFinder. Add the “RefFinder” in Line 21 Point 2: However, the transcriptome data was not given. The role of RNA-Seq should be clarified throughout the manuscript. It is not clear which transcriptome data they are referring to. If it is already published data it needs to be cited. If it is unpublished data they should give more information about it. Also, they should give more information about how they used the transcriptome data to select the genes. Reponse 2: In the introduction, transcriptome data of leaves and roots in L. rotata from five regions (GL, XW, YS, ZD, and CD regions, Figure 1 and Table 1) through Illumina high-throughput sequencing platform (Biomaker Technologies Company, Beijing, China). The cost of sequencing was funded by our project “the Department of Science and Technology of Qinghai Provincial (2021-ZJ-729 to F.Q.). The sequence analysis data of L. rotata transcriptome will appear in another article (Writing, unpublished), so this article does not present the detailed results of L. rotata transcriptome data. Point 3: The authors used geNorm, NormFinder, BestKeeper, and RefFinder, however, they do not mention about RefFinder in Lines 22, 145, and 373, and they talk about 3 analytical tools in Lines 369-379. Please provide results for all the tools used in the study. Reponse 3: In line 21, supplement “BestKeeper and RefFinder”. In line 157, 410 supplement “RefFinder”. In Lines 420-421, supplement the results of RefFinder”. Point 4: In Line 148, does “minimum Ct value” mean “median Ct value”? Reponse 4: “minimum Ct value” means the smallest Ct value, not “median Ct value” Point 5: Under title 4.1, various literature were given however, they are not related to the existing study. This title can be removed completely. The Discussion section should be completely rewritten. Reponse 5: Delete all subtitles of discussion. We modify the discussion section and add the last paragrdph in Line 440-458.Reviewer 3 Report
Dear Authors
Reviewer suggestion: Minor revision
Overall, the manuscript provides a nice study in Evaluation of Candidate Reference Genes for Gene Expression Analysis in Wild Lamiophlomis rotata as a Chinese and Tibetan Medicine Plant from Different-Altitude Habitats.
. However, there are some aspects that can be improved.
1. The introduction could be improved and more focused.
2. This reviewer thinks that an additional section in relation to novel aspects of this work.
- please add the below reference in the text:
Encapsulation of Plant Biocontrol Bacteria with Alginate as a Main Polymer Material
Finally, I will suggest this manuscript for publication after minor revision.
Author Response
Author's Reply to the Review Report (Reviewer 3) Point 1: The introduction could be improved and more focused. Response 1: The introduction section was modified and adjusted the paragraphs. Point 2. This reviewer thinks that an additional section in relation to novel aspects of this work. - please add the below reference in the text: Encapsulation of Plant Biocontrol Bacteria with Alginate as a Main Polymer Material Finally, I will suggest this manuscript for publication after minor revision. Response 2: Add the below reference 31 “Encapsulation of Plant Biocontrol Bacteria with Alginate as a Main Polymer Material “ in the introduction.Reviewer 4 Report
The manuscript entitled “Evaluation of Candidate Reference Genes for Gene Expression Analysis in Wild Lamiophlomis rotata as a Chinese and Tibetan Medicine Plant from Different-Altitude Habitats” by Wang et al. aimed to identify the most suitable internal reference genes in the leaves and roots of L. rotata from five different regions. Based on the RNA-Seq data of wild L. rotata from five different regions, the stability of 15 candidate internal reference genes was analyzed using different methods. They have identified TFIIS, EF-1α, and CYP22 as the most suitable internal reference genes in the leaves of L. rotata and OBP, TFIIS, and CYP22 as the optimal reference genes in the roots of L. rotata. The research work is interesting and the authors have data to support their hypothesis.
However, I have the following comments for the authors to improve the standard of the manuscript:
1. In the Introduction line 75-80, the authors have mentioned, “In this study, 15 candidate internal reference genes were screened on the basis of transcriptome data from L. rotata leaves and roots. These genes were actin 8 (ACT8), cyclophilin 22 (CYP22), cyclophilin 95 (CYP95), TIP 41-like protein (TIP41), translation elonga-77 tion factor (TFIIS), EF-1 α, actin 7 (ACT7), protein phosphatase 2A (PP2A), oxysterol-binding protein (OBP), tubulin beta chain (TUB), cyclophilin23 (CYP23), ribochemical protein large subunit (RPL), anthranilate phosphoribosyltransferase (TrpD), acetylcholine deacetylase (AO), and alpha colloid NSF attachment protein (SNAP). However they have not discussed anywhere in the manuscript regarding the importance of selecting these 15 genes.
2. Discussion needs major revision and in a more systematic way. I think the authors need to explain regarding the selection of these 15 genes and their importance briefly. Then they can discuss regarding the advantage of using different methods and the genes that they have identified in leaves and roots in L. rotata from five different regions. They can provide references to support their findings. However, the authors have discussed more on others findings in the manuscript.
3. In the Introduction line 42-46, the authors have mentioned, “Research on the biosynthetic pathway of iridoid, flavonoid, and phenylethanol glycoside compounds in L. rotata has not been reported. The expression pattern of key genes in the biosynthetic pathway of secondary metabolism in L. rotata must be studied, and the research results could provide clues for the molecular mechanism of pharmaceutical ingredient accumulation in L. rotata.” Are these 15 genes related to the biosynthetic pathway of iridoid, flavonoid, and phenylethanol glycoside compounds? If the identified genes in leaves and roots are related to any of these secondary metabolic pathways, please discuss about them based on your findings.
Author Response
Author's Reply to the Review Report (Reviewer 4) Point 1: In the Introduction line 75-80, the authors have mentioned, “In this study, 15 candidate internal reference genes were screened on the basis of transcriptome data from L. rotata leaves and roots. These genes were actin 8 (ACT8), cyclophilin 22 (CYP22), cyclophilin 95 (CYP95), TIP 41-like protein (TIP41), translation elonga-77 tion factor (TFIIS), EF-1 α, actin 7 (ACT7), protein phosphatase 2A (PP2A), oxysterol-binding protein (OBP), tubulin beta chain (TUB), cyclophilin23 (CYP23), ribochemical protein large subunit (RPL), anthranilate phosphoribosyltransferase (TrpD), acetylcholine deacetylase (AO), and alpha colloid NSF attachment protein (SNAP). However they have not discussed anywhere in the manuscript regarding the importance of selecting these 15 genes. Response 1: Add contents in lines 427-433 in discussion. Point 2: Discussion needs major revision and in a more systematic way. I think the authors need to explain regarding the selection of these 15 genes and their importance briefly. Then they can discuss regarding the advantage of using different methods and the genes that they have identified in leaves and roots in L. rotata from five different regions. They can provide references to support their findings. However, the authors have discussed more on others findings in the manuscript. Response 2: Add contents in lines 427-433 in discussion. Add the last paragraph in lines 440-458 in discussion. Point 3: In the Introduction line 42-46, the authors have mentioned, “Research on the biosynthetic pathway of iridoid, flavonoid, and phenylethanol glycoside compounds in L. rotata has not been reported. The expression pattern of key genes in the biosynthetic pathway of secondary metabolism in L. rotata must be studied, and the research results could provide clues for the molecular mechanism of pharmaceutical ingredient accumulation in L. rotata.” Are these 15 genes related to the biosynthetic pathway of iridoid, flavonoid, and phenylethanol glycoside compounds? If the identified genes in leaves and roots are related to any of these secondary metabolic pathways, please discuss about them based on your findings. Response 3: Although these 15 reference genes were not related to the biosynthetic pathway of iridoid, flavonoid, and phenylethanol glycoside compounds, the proper or stable reference genes among 15 genes were screened by RT-qPCR. Further, the proper or stable reference genes were used for analyzing the expressions of pecific genes related to biosynthetic pathway of secondary metabolism in L. rotata. We add the last paragraph in lines 440-458 in discussion. Add the sentence in conclusion and Future Perspectives in lines 470-474.Round 2
Reviewer 2 Report
The authors provide a response for each comment. However, for the source of transcriptome data, no evidence was provided. I suggest that the transcriptome data can be stored in a public database then it can be used it.
Reviewer 4 Report
I do not have anymore comments for the authors.